# Single versus dual antiplatelet therapy following peripheral arterial endovascular intervention for chronic limb threatening ischaemia: Retrospective cohort study

**Natasha Chinai** [1], **Graeme K. Ambler**[1,2], **Bethany G. Wardle**[1], **Dafydd Locker**[3], **Dave Bosanquet** [3], **Nimit Goyal**[4], **Christopher Chick**[4], **Robert J. Hinchliffe**[1,2], **Christopher P. Twine**[1,2]*

1 Department of Vascular Surgery, Southmead Hospital, Bristol, England, United Kingdom, 2 Bristol Centre for Surgical Research, Bristol Medical School, University of Bristol, Bristol, England, United Kingdom, 3 Department of Vascular Surgery, Royal Gwent Hospital, Newport, Wales, United Kingdom, 4 Department of Radiology, Royal Gwent Hospital, Newport, Wales, United Kingdom

* chris_twine@hotmail.com

**Data Availability Statement:** Data cannot be shared publicly as per the standard approvals from Research and Development at the Aneurin Bevan

## Abstract

### Objectives

Antiplatelet therapy following peripheral arterial endovascular intervention lacks high quality evidence to guide practice. The aim of this study was to assess the effect of three months of dual antiplatelet therapy on amputation-free survival following peripheral arterial endovascular intervention in patients with chronic limb threatening ischemia.

### Methods

A retrospective review of symptomatic patients undergoing primary peripheral arterial endovascular intervention over a seven-year period was performed. The primary outcome measure was amputation-free survival. A sample size calculation based on previous cohort studies suggested that 629 limbs would be required to show a difference between single and dual therapy. Kaplan-Meier estimates and multivariate logistic regression analysis of recorded baseline characteristics was performed to determine predictors of amputation-free survival. Dual antiplatelet therapy was routinely given for 3 months.

### Results

754 limbs were treated with primary angioplasty and/or stenting over a 7-year period, 508 of these for chronic limb threatening ischemia. There was no difference in unadjusted amputation-free survival between patients with chronic limb threatening ischaemia taking single vs. dual antiplatelet therapy (69% vs. 74% respectively Log rank Chi$^2$ = 0.1, $p$ = .72). After adjusting for confounders, at 1 year there was also no significant difference in amputation-free survival between patients taking single vs. dual antiplatelet therapy [OR 0.8, 95% CI 0.5–1.2, $p$ = .3]. There was no difference in rates of major bleeding between single and dual antiplatelet therapy.

University Health Board. Data are available from the South East Wales (Gwent) Ethics Committee for researchers who meet the criteria for access to confidential data. The project was registered and approved by Research and Development at Aneurin Bevan University Health board: St Woolos Hospital, 131 Stow Hill, Newport NP20 4SZ (Data requests from future researchers should be sent to email: ABB.R&D@wales.nhs.uk).

**Funding:** This study was supported by the NIHR Biomedical Research Centre at University Hospitals Bristol NHS Foundation Trust and the University of Bristol. The views expressed in this publication are those of the authors and not necessarily those of the NHS, the National Institute for Health Research or the Department of Health and Social Care. GKA was supported by a Royal College of Surgeons Research Fellowship. CPT was supported by Learning and Research at North Bristol NHS trust.

**Competing interests:** CPT and 23 RJH sit on the European Society for Vascular Surgery (ESVS) guidelines committee. CPT is chairing the development of the ESVS antithrombotic guideline for peripheral arterial disease.

## Conclusions

There was no clear evidence of reduced amputation-free survival in patients with chronic limb threatening ischemia undergoing peripheral arterial endovascular intervention being treated with dual antiplatelet therapy for 3 months. This is at odds with other retrospective case series and highlights the limitations in basing clinical practice on such data. There is a need for an adequately powered, independent randomised trial to definitively answer the question.

## Introduction

Antiplatelet therapy following peripheral arterial endovascular intervention lacks high-quality evidence to guide practice [1]. This is surprising given the large number of randomised trials to guide antiplatelet practice in other areas of treatment for peripheral arterial disease, and causes conflict between recommendations in guidelines [2,3]. It has also resulted in a variation in clinical practice between clinicians, centers and countries [4].

There is one small randomised trial examining single vs. dual antiplatelet therapy after endovascular intervention [5,6]. This lacked the power to compare clinically relevant outcomes such as amputation rates or amputation-free survival. The trial recruited forty patients in each arm and at 12 months there was no statistically significant difference between target lesion revascularisation in the dual antiplatelet arm (Aspirin and Clopidogrel) over Aspirin single therapy.

There is case-series evidence to support a benefit of dual antiplatelet therapy in patients undergoing peripheral endovascular intervention [7–9]. However, these series included a heterogenous patient mix including patients undergoing intervention for claudication and patients undergoing secondary intervention. The most important group of patients undergoing intervention for peripheral arterial disease are those with chronic limb threatening ischaemia who are at the highest risk of amputation or death if treatment fails [10]. This group is vastly under-represented in previous case series. Moreover, long (over 6 months) durations of antiplatelet therapy was used in these studies, however clinical practice varies worldwide and many UK units, including ours, used short (less than 6 months) courses [1,11].

The aim of this paper was to examine the clinically relevant differences between single and dual antiplatelet therapy following peripheral arterial intervention for chronic limb threatening ischaemia in a way which minimises previous case series' flaws: a powered, clearly specified cohort with statistically valid follow up criteria and adjusted outcome reporting.

## Methods

### Study design

This was a retrospective cohort study of patients undergoing primary lower limb peripheral arterial endovascular intervention for claudication or chronic limb threatening ischaemia. The study was approved by the Aneurin Bevan University Health Board Research and Development department (ABUHB R&D reference number: SA/710/16) and registered on ANZCTR. The research and development approval waives the requirement for informed consent which is in place for all approved studies using the clinical workstation browser. The study is reported in line with the STROBE statement [12].

## Study patients

Patients undergoing primary lower limb arterial endovascular intervention at the Royal Gwent Hospital, Newport between January 2010 and January 2017 were screened for inclusion. The Health Board's electronic clinical workstation browser was used to collect anonymized data; all data were anonymized for collection and analysis. Clinical workstation is linked to the office for national statistics in the UK for mortality reporting.

Pre-procedural data associated with poor amputation-free survival [13,14] was collected on each patient including baseline blood tests (white cell count, haemoglobin, platelet count, urea, creatinine, eGFR and albumin), co-morbidities (ischaemic heart disease, diabetes, hypertension, congestive cardiac failure, chronic obstructive airways disease and cerebrovascular accident), smoking status, angiotensin converting enzyme inhibitor or angiotensin II receptor blocker and statin usage. Data on anatomical location of the lesion and endovascular treatment device ((balloon—plain or drug eluting; stent—bare metal, drug eluting or covered) was also collected.

All patients were discussed by a multidisciplinary team before intervention. After discharge following intervention patients were routinely followed up at 6 weeks in an outpatient clinic. Patients with tissue loss were followed up in a wound healing service until the point of healing (defined as full epithelialization). Patients undergoing stent implantation were followed up with duplex surveillance at 3, 6, 9 and 12 months. Recurrent stenoses >75% were discussed by a multidisciplinary team and offered repeat intervention if clinically appropriate. The study end date was 30/09/2018, however, survival and re-intervention times were defined up to last known date of follow-up. The follow-up period was measured relative to the declared study end date. A follow-up index was calculated for each patient. This is defined as the ratio between the investigated follow-up period and the theoretically possible follow-up period to the pre-specified study end date [15]. The international classification of diseases was used for disease definitions, apart from ischaemic heart disease which includes angina and myocardial infarction definitions [16].

## Inclusion criteria

- Primary (first) lower limb intervention for atherosclerotic peripheral arterial disease

- Angioplasty or stenting with any device

- Angioplasty or stenting of the distal infra-renal aorta with common iliac disease; common or external iliac arteries; superficial femoral artery; popliteal artery; tibio-peroneal trunk; anterior or posterior tibial artery; peroneal artery or any pedal artery.

## Exclusion criteria

- Patients undergoing treatment for aneurysmal disease or for a complication of treatment of aneurysmal disease

- Secondary or endovascular re-intervention to the same limb

- Treatment proximal to the infra-renal aorta; mesenteric or renal vessels; upper limb or head and neck vessels

- Hybrid revascularization procedures

- Patients undergoing intra-arterial embolectomy, thrombectomy or thrombolysis, other than those where this was commenced immediately following primary angioplasty or stenting in order to treat a complication

- Venous procedures (e.g. deep venous stenting)

- Angioplasty or stenting of arterio-venous fistulas or arterio-venous malformations

- Patients receiving therapeutic anticoagulation

- Patients receiving dual antiplatelet therapy for another cause prior to peripheral endovascular intervention

- Patients receiving no anti-platelet therapy at the time of discharge

## Endpoints

**Primary endpoint.**   Amputation-free survival (composite of mortality plus major lower limb amputation defined as above ankle) [2]. We have performed patient focus group work as part of the run up for a randomised trial of single vs. dual antiplatelet therapy which confirmed this to be the most important patient centered outcome after treatment for chronic limb threatening ischaemia. It is also a widely accepted effectiveness outcome in chronic limb threatening ischaemia used in several major randomised trials [2].

**Primary safety endpoint.**   Major bleeding (using the International Society on Thrombosis and Haemostasis definition) [17].

**Secondary endpoints.**   Survival, limb salvage, target lesion revascularisation, trends in dual antiplatelet therapy prescription with time.

## Antiplatelet therapy

Antiplatelet therapy was prescribed by the consultant vascular surgeon or interventional radiologist looking after the patient. Choice of antiplatelet therapy was at the discretion of the attending surgeon/radiologist. There were no specific criteria for using single or dual antiplatelet therapy, it was dictated by the practice of the person performing the procedure. The hospital prescription would last for 4–6 weeks and a summary would be given to the patient and GP for continuation. Dual antiplatelet therapy was continued for a period of 3 months after intervention and long term single antiplatelet therapy was given after this.

## Sample size calculation

In the previously published case series the rate of dual antiplatelet therapy varied: 31% in a Swedish registry study [7], 55% in a single centre study from America [8] and 69% in an American registry study [9]. The benefits of dual therapy also varied between the studies; with a hazard ratio [HR 0.54] in favour of dual antiplatelet therapy in the Armstrong *et al* single centre study, HR 0.89 for mortality in the American registry, and HR 0.72 and HR 0.77 for survival and amputation respectively in the Swedish registry. The variability in previous similar case series made it difficult to estimate the prevalence and treatment effect of dual therapy in our cohort *a-priori*.

Patients with chronic limb threatening ischaemia could not be analysed separately for a sample size calculation from the cohort studies above. Therefore a total sample size calculation was performed for all patients undergoing intervention presuming a similar case mix and the chronic limb threatening ischaemia patients were analysed alone in our study. Sample size

calculations were based on a 40% prevalence of dual antiplatelet therapy and a hazard ratio 0.65 to estimate the treatment effect of dual therapy (our primary outcome is amputation-free survival) in our cohort. With mortality rates between 20%-30% and amputation rates 8%-15% in the previous three case series, the estimated event rate was 28%. Therefore, using standard size sample size calculation [18], we estimated that six hundred and twenty nine patients would be required to have 80% power to show a difference comparable to the published literature at the 5% level with a median follow-up of 36 months. Based on previous audit we knew that 120–140 procedures were performed annually, so on the assumption that we were likely to exclude approximately 25% of patients due to treatment of patients without chronic limb threatening ischaemia or who were on anticoagulation, we estimated that seven years (the period during which electronic records were available) should be enough to achieve adequate power to address the study question.

## Statistical analysis

All data was anonymized prior to analysis. The data were cleaned by first resolving transcriptional discrepancies and clinical conflicts. Aberrant and extreme values were removed or transformed if the cause was inconsistent measurement units. All variables missing more than 15% of data were excluded from analyses. The sample median was substituted for missing continuous variables and the mode for missing categorical variables.

For descriptive data with a normal distribution the mean±standard deviation (SD) was specified and comparisons were performed using a two-sided student's t-test. For data with non-normal distribution the median and interquartile range were given and comparisons made using the Mann Whitney U test. For categorical variables comparisons were performed using the Fisher's exact test. Unadjusted event free survival was estimated using the Kaplan-Meier method and between group difference assessed using the log-rank test [19] Potential confounders of amputation-free survival were adjusted for: Baseline white cell count, haemoglobin, platelet count, urea, creatinine, eGFR, albumin, ischaemic heart disease, hypertension, diabetes mellitus, cerebrovascular event, congestive cardiac failure, chronic obstructive airways disease, statin, angiotensin converting enzyme inhibitor or angiotensin II receptor blocker usage, anatomical location of the lesion and treating endovascular device (balloon—plain or drug eluting; stent—bare metal, drug eluting or covered). Confounder correction using Cox Proportional Hazards method was undertaken with stepwise selection of confounders through minimization of the Akaike Information Criterion to reduce model over-fitting. Scaled Schoenfeld residuals were analysed to assess for violation of the proportional hazards assumption [20].

The primary endpoint was also examined at 1-year post intervention as trials in cardiology suggest the greatest benefit to this time period [21], and to minimize error due to attrition. We used logistic regression analysis and odds ratios to analyze risk factors associated with poor amputation-free survival. Multivariate logistic regression was used to adjust for confounders (baseline white cell count, haemoglobin, platelet count, urea, creatinine, eGFR, albumin, ischaemic heart disease, hypertension, diabetes mellitus, cerebrovascular event, congestive cardiac failure, chronic obstructive airways disease, statin, angiotensin converting enzyme inhibitor or angiotensin II receptor blocker usage, anatomical location of the lesion and treating endovascular device), with stepwise selection of significant confounding variables by minimizing the Akaike information criterion (AIC) [22]. A further sensitivity analysis was performed to assess the effect of drug eluting devices. Linear regression was carried out to examine trends of antiplatelet prescription with time. All statistical analysis was performed within the R

statistical programming environment version 3.5.1 and for all tests a $p$ value of $< .05$ was considered significant.

## Results

### Patient demographics and interventions

Nine hundred and twenty-seven patients were screened for inclusion in the study (Fig 1). Six hundred and twenty-five patients and seven hundred and fifty-four limbs met the inclusion/exclusion criteria were treated with primary angioplasty or stenting over the seven-year period. Five hundred and eight of these were performed in patients with chronic limb threatening ischaemia. The mean age of the entire cohort was 69.8±10.9 years; three hundred and ninety-six (63%) patients were male. The median follow-up time was 29 (8–33) months, with a mean follow-up index of 0.94±0.1. There were 244 deaths during the study period (208 patients with chronic limb threatening ischaemia and 36 with intermittent claudication). Only two variables were excluded because of >15% missing data: smoking and the Wound Ischaemia and Foot infection score.

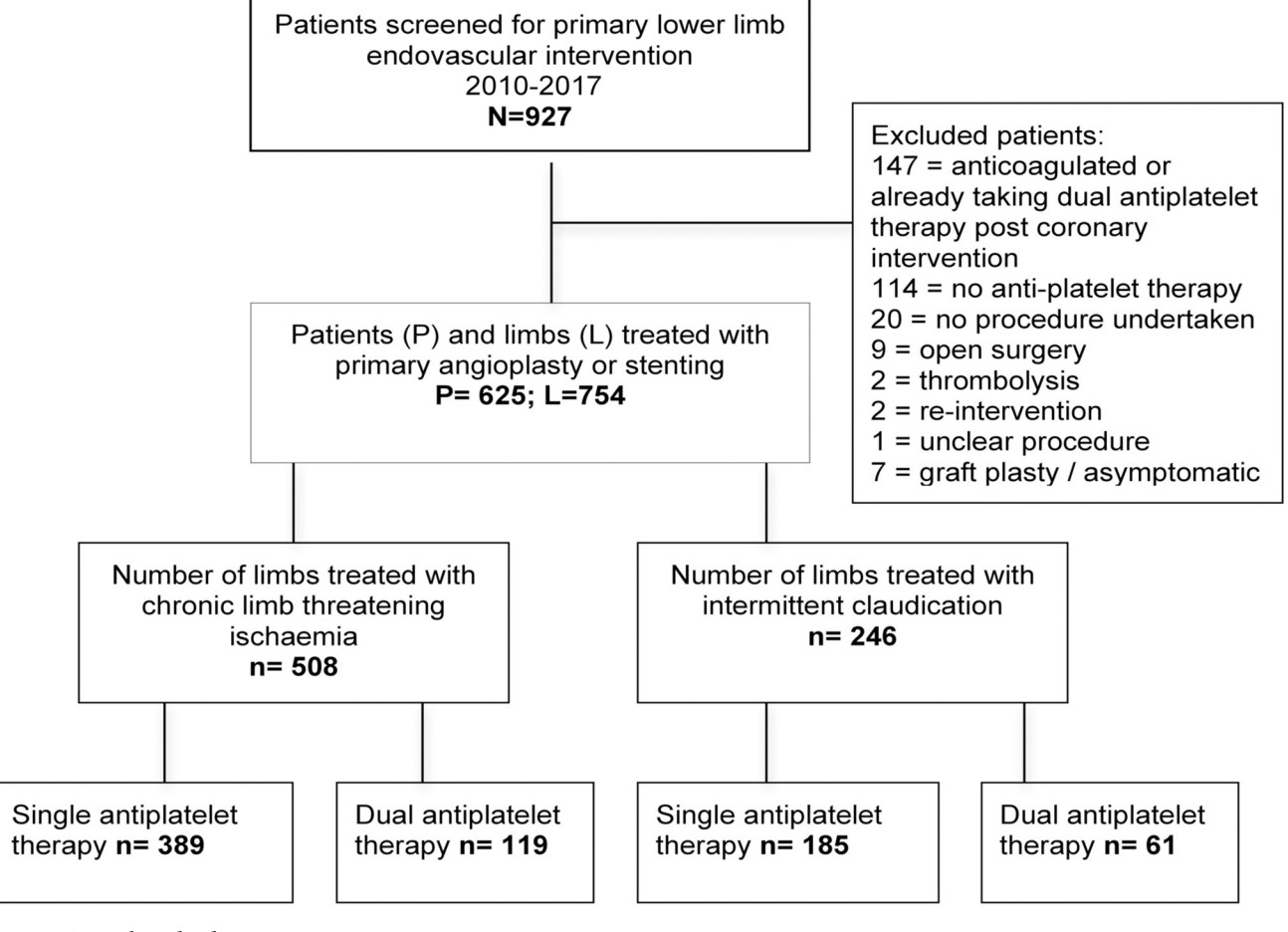

**Fig 1. Patient cohort development.**

## Amputation-free survival in patients with chronic limb threatening ischaemia

There were differences in single and dual therapy by anatomical location and endovascular treatment device used (Table 1).

Overall unadjusted amputation-free survival for patients with chronic limb threatening ischaemia was 79% at 6 months, 71% at one year, 63% at 2 years and 55% at 3 years. There was no difference in unadjusted amputation-free survival (Log rank $Chi^2 = 0.1$, $p = .72$ Fig 2).

On adjusting for confounders (baseline white cell count, haemoglobin, platelet count, urea, creatinine, eGFR, albumin, ischaemic heart disease, hypertension, diabetes mellitus, cerebrovascular event, congestive cardiac failure, chronic obstructive airways disease, statin, angiotensin converting enzyme inhibitor or angiotensin II receptor blocker usage, anatomical location of the lesion and endovascular device, S1 Table), amputation-free survival was not associated with choice of antiplatelet therapy [HR 0.94, 95% CI 0.7–1.3, $p = .7$].

The cardiology literature suggests benefit to one year from a course of dual antiplatelet therapy.[11] When truncating follow-up at 1 year in this cohort there was also no significant difference in amputation-free survival between patients taking single or dual antiplatelet therapy [OR 0.8, 95% CI 0.5–1.2, $p = .3$] after adjusting for the following confounding variables: baseline white cell count, haemoglobin, platelet count, urea, creatinine, eGFR, albumin, ischaemic heart disease, hypertension, diabetes mellitus, cerebrovascular event, congestive cardiac failure, chronic obstructive airways disease, statin, angiotensin converting enzyme inhibitor or angiotensin II receptor blocker usage, anatomical location of the lesion and endovascular device (S1 Table).

## Major bleeding

There was no statistically significant difference in major bleeding between the dual antiplatelet therapy (n = 8) and single therapy group (n = 26) of patients ($p = 1$). There was no statistically significant difference in major bleeding between aspirin and clopidogrel in the single therapy group (21/437 patients taking aspirin vs 5/145 patients taking clopidogrel, $p = 0.6$). These trends were the same for chronic limb threatening ischaemia patients alone ($p = .82$).

## Secondary endpoints in patients with chronic limb threatening ischaemia

Survival was no different between patients receiving single and dual antiplatelet therapy (Log Rank Chi2 = 0, $p = .9$). This was also the case after adjusting for confounders (baseline white cell count, haemoglobin, platelet count, urea, creatinine, eGFR, albumin, ischaemic heart disease, hypertension, diabetes mellitus, cerebrovascular event, congestive cardiac failure, chronic obstructive airways disease, statin, angiotensin converting enzyme inhibitor or angiotensin II receptor blocker usage, anatomical location of the lesion and endovascular device, S1 Table; HR 1.0, 95% CI 0.7–1.4, $p = .9$).

There were eighty-five post intervention major lower limb amputations: 70 (17.6%) in the single therapy group and 15 (12.6%) in the dual therapy group. There was no difference in limb salvage rates between patients receiving single or dual antiplatelet therapy (Log Rank Chi2 = 1.7, $p = .2$). This was also the case after adjusting for confounders (baseline white cell count, haemoglobin, platelet count, urea, creatinine, eGFR, albumin, ischaemic heart disease, hypertension, diabetes mellitus, cerebrovascular event, congestive cardiac failure, chronic obstructive airways disease, statin and angiotensin converting enzyme inhibitor or angiotensin II receptor blocker usage, anatomical location of the lesion and endovascular device, S1 Table; HR 0.6, 95% CI 0.3–1.1, $p = .1$).

**Table 1. Baseline characteristics of patients with chronic limb threatening ischaemia.**

| | Single therapy n = 389 | | Dual Therapy n = 119 | P |
|---|---|---|---|---|
| **Aspirin (n)** | 318 | **Aspirin + Clopidogrel** | 113 | - |
| **Clopidogrel (n)** | 71 | **Aspirin + Prasugrel** | 5 | - |
| | | **Aspirin + Ticagrelor** | 1 | - |
| **Concurrent statin n(%)** | 285(73) | | 92(77) | 0.29 |
| **Concurrent ACEi n(%)** | 200(51) | | 59(50) | 0.92 |
| **Co-morbidities** | | | | |
| IHD **n(%)**[*] | 173((44) | | 51(43) | 0.92 |
| Diabetes **n(%)** | 222(57) | | 71(60) | 0.53 |
| Hypertension **n(%)** | 303(78) | | 97(82) | 0.26 |
| CVE **n(%)**[**] | 55(14) | | 25(21) | 0.06 |
| COPD **n(%)** | 66(17) | | 22(18) | 0.68 |
| CCF **n(%)** | 83(21) | | 30(25) | 0.32 |
| **Blood results (Median(IQR))** | | | | |
| Haemoglobin (g/L) | 122 (109–138) | | 120 (108–136) | 0.18 |
| Platelets ($10^9$/L) | 291 (236–365) | | 293 (229–364) | 0.16 |
| WCC ($10^9$/L) | 9.4 (7.6–11.4) | | 9.4 (7.6–11.4) | 0.27 |
| Urea (mmol/L) | 6.0 (4.4–8.1) | | 5.9 (4.4–8.3) | 0.22 |
| Creatinine (mmol/L) | 79 (67–106) | | 79 (68–108) | 0.94 |
| eGFR | 76 (57–98) | | 77 (56–99) | 0.4 |
| Albumin (g/L) | 33 (28–37) | | 32 (27–36) | 0.21 |
| **Minor amputations (n)** | 57 | | 19 | 0.66 |
| **Major amputations** | | | | |
| Transtibial (n) | 45 | | 9 | 0.24 |
| Transfemoral (n) | 25 | | 6 | 0.6 |
| **Major bleeding events (n)** | 18 | | 6 | 0.82 |
| **Anatomical location** | | | | |
| Aorto-iliac | 102 | | 57 | <0.00001 |
| Femoro-popliteal[a] | 126 | | 41 | 0.74 |
| Below the knee[b] | 88 | | 9 | 0.000245 |
| Iliac + Fem-pop | 6 | | 1 | 1 |
| Iliac + BTK | 1 | | 0 | 1 |
| Fem-pop + BTK | 65 | | 11 | 0.06 |
| **Endovascular device** | | | | |
| Balloon angioplasty | 261 | | 50 | <0.00001 |
| Drug eluting balloon | 98 | | 55 | 0.000012 |
| Bare metal stent | 5 | | 2 | 0.67 |
| Drug eluting stent | 5 | | 6 | 0.013951 |
| Covered stent | 1 | | 4 | 0.003138 |

IHD = Ischaemic Heart Disease; CVE = Cerebrovascular Event; COPD = Chronic Obstructive Pulmonary Disease; CCF = Congestive Cardiac Failure; WCC = White Cell Count

[*] Ischaemic heart disease was defined as angina and/or previous myocardial infarction.

[**] CVE includes both stroke and transient ischaemic attack

[a] Femoro-popliteal (fem-pop) ends at the level of P2

[b] Below the knee (BTK) starts at P3

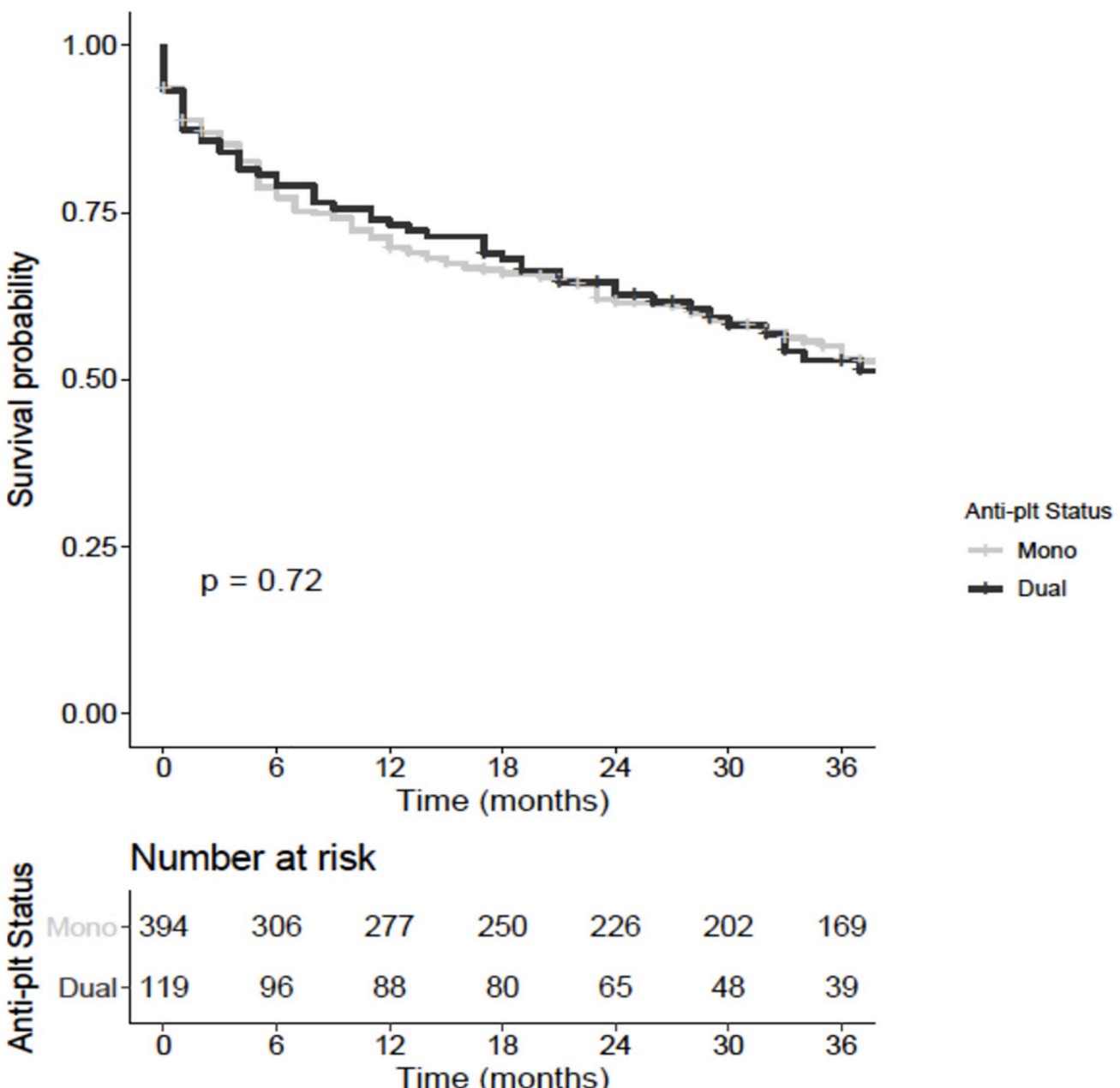

**Fig 2. Kaplan-Meir curve showing amputation-free survival for patients with chronic limb threatening ischaemia receiving single or dual antiplatelet therapy following peripheral endovascular intervention.** The Y axis shows probability of survival and the X axis shows time in months.

Target lesion revascularisation rates were no different between the groups. Re-intervention was performed in 152 (29%) limbs with 120 (31%) on single therapy and 32 (27%) on dual antiplatelet therapy post primary intervention ($p$ = .5). The median time to reintervention was 6 (2–18) months.

A change in antiplatelet prescription was noted over time with clinicians prescribing clopidogrel rather than aspirin single therapy and dual antiplatelet therapy more frequently with

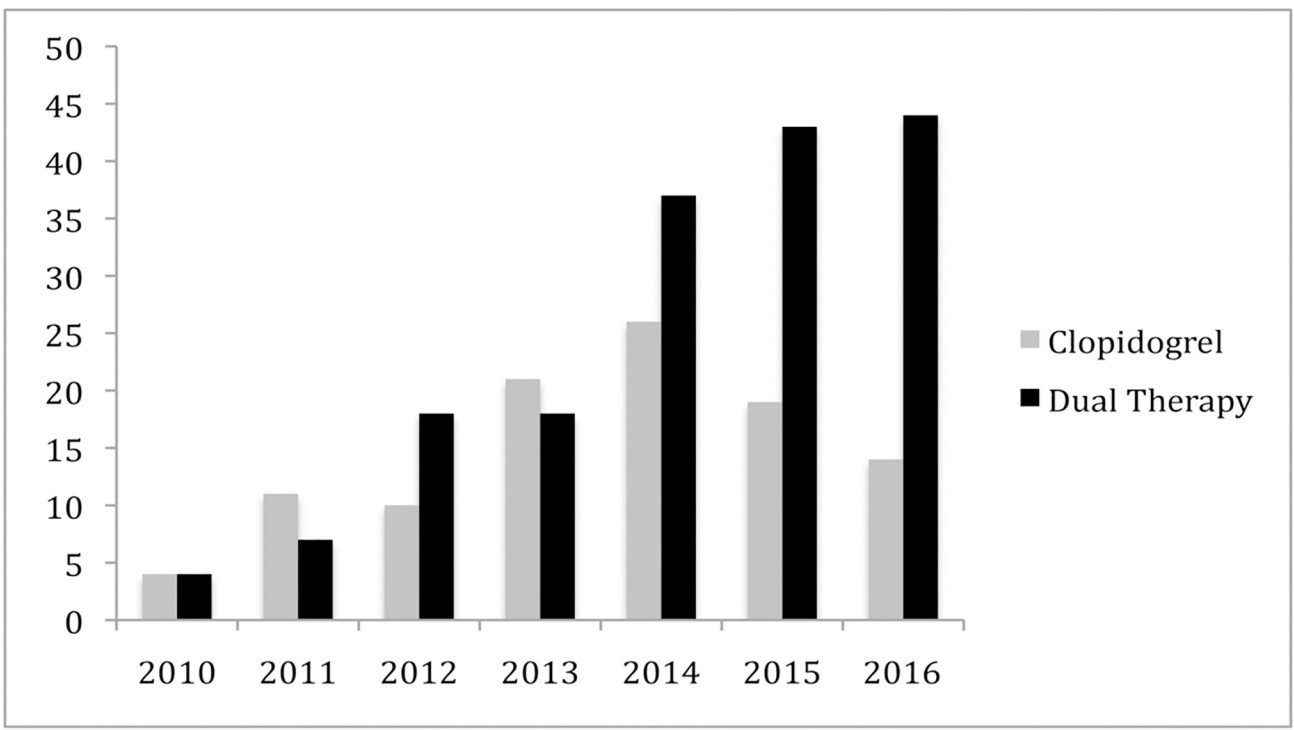

**Fig 3. Change in antiplatelet prescribing post peripheral endovascular intervention over time.** Dual antiplatelet therapy ($R^2$ = 0.94, $p$ = .0003) Clopidogrel ($R^2$ = 0.41, $p$ = .12). Y axis shows number of cases and X axis shows year of prescription.

time (Fig 3). There was an increasing trend in the proportion of patients being discharged on dual antiplatelet therapy increased from 2% in 2010 to 26% in 2016 ($p$ = .0003).

## Discussion

There was no statistically significant difference in amputation-free survival or major bleeding events between patients on single or dual antiplatelet therapy after lower limb endovascular intervention for chronic limb threatening ischaemia.

The main finding is interesting for a number of reasons. Although the study had appropriate statistical power to detect differences similar to the previously published literature, the case mix is different with fewer diabetics in our study than one of the large cohort studies used for the calculation [7]. As event rates were higher in the diabetic cohort in this study it may be no effect is shown in our data as a result. There were difference in use of single and dual therapy by both anatomical location and endovascular device used. This is in keeping with a previous international survey of practice [4] and would potentially confound previous unadjusted studies.

Three months of dual therapy was used in our cohort as opposed to longer courses in previous cohorts [8,9]. In trials comparing the duration of dual antiplatelet therapy after percutaneous coronary intervention there are debates over the optimal length of treatment because of the balance of risk and benefit [11]. Durations of antiplatelet therapy longer than twelve months after percutaneous coronary stenting cause harm as a result of increased major bleeding rates which means most guidelines do not recommend courses over six to twelve months [23–26]. Conversely, the STOPDAPT-2 study showed benefit for only a 1 month course of dual therapy [27]. As this is undefined following percutaneous lower limb intervention it may

be the case that six or twelve months rather than three months of dual therapy would show a difference between the groups at one year.

Major bleeding events showed no difference between the groups. The overall rate of major bleeding was 54 per 1000 patients which is similar to that reported in previous randomised trials [1]. The sample size in this study is therefore too small to detect a significant difference, but it is reassuring that there is no large discrepancy.

The trend in increasing use of dual therapy over time is in keeping with previous work showing this to be the case [4]. It may be that clinical practice is being extrapolated from cardiology or from randomised trials of newer technologies such as drug eluting stents, where dual therapy was often specified in the protocol with no justification [28]. Either way there is a potential that this causes more harm than good while we do not fully understand the benefit for amputation-free survival.

This analysis has a number of limitations. Retrospective studies are limited by the type and quality of the data collected. Even though a sample size calculation was performed this is not as powerful for retrospective data as if the study was prospective. It is helpful to guide the number of patients included in the study but cannot be seen as definitive. There is still the possibility of type 2 error in any of the comparison, especially as the groups become smaller. Smoking status is associated with increasing atherosclerotic burden as well as increased risk of major adverse cardiovascular events [29]. There was lack of sufficient data on the smoking status of patients in this cohort so we were unable to account for the effect of confounding due to this factor in our analysis. A lack of strict antiplatelet protocol means that longer or shorter durations may have been taken by patients and no compliance checks were made.

The study also has several major strengths for a retrospective cohort. A sample size calculation was performed based on previous publications in this area and there was sufficient statistical power to detect differences similar to those seen in previous published studies. We focused on patient centered clinical endpoints in patients with chronic limb threatening ischaemia, who benefit most from these interventions. Inclusion/exclusion and endpoint criteria were prespecified and adhered to and follow-up reporting was robust, as demonstrated by the mean follow-up index of 0.94.

This research alone is not high enough quality to make recommendations for practice, nor is any other cohort study or previously published randomized trial in the area [5–9]. The conflicting results make it clear that an adequately powered randomised trial in this area is required to make practice recommendations.

## Conclusion

There was no evidence of improved amputation-free survival in patients with chronic limb threatening ischaemia undergoing peripheral arterial endovascular intervention being treated with 3 months of dual antiplatelet therapy over those on a single agent. This is at odds with other retrospective case series and highlights the dangers in basing clinical practice on such data. There is a need for a high-quality randomised trial to definitively answer the question of whether dual antiplatelet therapy is of benefit after peripheral endovascular intervention.

## Supporting information

**S1 Table. Multivariate analysis with confounder correction.**
(DOCX)

## Author Contributions

**Conceptualization:** Graeme K. Ambler, Nimit Goyal, Christopher Chick, Christopher P. Twine.

**Data curation:** Natasha Chinai, Graeme K. Ambler.

**Formal analysis:** Natasha Chinai, Graeme K. Ambler.

**Investigation:** Bethany G. Wardle, Dafydd Locker, Dave Bosanquet.

**Methodology:** Natasha Chinai, Graeme K. Ambler, Christopher P. Twine.

**Project administration:** Graeme K. Ambler, Christopher P. Twine.

**Resources:** Natasha Chinai.

**Supervision:** Christopher P. Twine.

**Validation:** Graeme K. Ambler.

**Visualization:** Natasha Chinai, Christopher P. Twine.

**Writing – original draft:** Natasha Chinai.

**Writing – review & editing:** Graeme K. Ambler, Robert J. Hinchliffe, Christopher P. Twine.

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
