## [Decision Letter · Decision Letter 0]

24 Mar 2020

PONE-D-20-04980

Single versus dual antiplatelet therapy following peripheral arterial endovascular intervention for chronic limb threatening ischaemia: retrospective cohort study

PLOS ONE

Dear Mr Twine,

Thank you for submitting your manuscript to PLOS ONE. After careful consideration, we feel that it has merit but does not fully meet PLOS ONE’s publication criteria as it currently stands. Therefore, we invite you to submit a revised version of the manuscript that addresses the points raised during the review process.

We would appreciate receiving your revised manuscript by May 08 2020 11:59PM. To enhance the reproducibility of your results, we recommend that if applicable you deposit your laboratory protocols in protocols.io, where a protocol can be assigned its own identifier (DOI) such that it can be cited independently in the future. For instructions see: http://journals.plos.org/plosone/s/submission-guidelines#loc-laboratory-protocols

We look forward to receiving your revised manuscript.

Kind regards,

Athanasios Saratzis

Academic Editor

PLOS ONE

Journal Requirements:

2. In ethics statement in the manuscript and in the online submission form, please provide additional information about the patient records used in your retrospective study. Specifically, please ensure that you have discussed whether all data were fully anonymized before you accessed them and/or whether the IRB or ethics committee waived the requirement for informed consent. If patients provided informed written consent to have data from their medical records used in research, please include this information.

"This study was supported by the NIHR Biomedical Research Centre at

432 University Hospitals Bristol NHS Foundation Trust and the University of

433 Bristol. The views expressed in this publication are those of the authors and

434 not necessarily those of the NHS, the National Institute for Health Research or

435 the Department of Health and Social Care. GKA was supported by a Royal

436 College of Surgeons Research Fellowship. CPT was supported by Learning

437 and Research at North Bristol NHS trust.".

i) We note that you have provided funding information that is not currently declared in your Funding Statement. However, funding information should not appear in the Acknowledgments section or other areas of your manuscript. We will only publish funding information present in the Funding Statement section of the online submission form.

ii) Please remove any funding-related text from the manuscript and let us know how you would like to update your Funding Statement. Currently, your Funding Statement reads as follows:

"The author(s) received no specific funding for this work.".

Additional Editor Comments (if provided):

Dear Chris and Rob,

Thank you very much for your submission. Some comments were raised by the Reviewers. Please note that 3 Reviewers have kindly provided comments - the 3rd Reviewer's comments can be accessed on the attached filed as they had issues with using the online system. I hope that is OK. Please contact me directly if you have problems accessing that attachment.

Thank you very much for your submission.

Reviewers' comments:

Reviewer's Responses to Questions

**Comments to the Author**

1. Is the manuscript technically sound, and do the data support the conclusions?

Reviewer #1: Yes

Reviewer #2: Yes

2. Has the statistical analysis been performed appropriately and rigorously? 

Reviewer #1: No

Reviewer #2: Yes

3. Have the authors made all data underlying the findings in their manuscript fully available?

Reviewer #1: No

Reviewer #2: Yes

4. Is the manuscript presented in an intelligible fashion and written in standard English?

Reviewer #1: Yes

Reviewer #2: Yes

5. Review Comments to the Author

Reviewer #1: General comments: The authors present a case controlled study on the possible association between DAPT and AFS. I have several comments outlined below.

1. I wonder what the reason was to study AFS as your primary endpoint because this would reduce the “power” of your study when looking at isolated limb outcomes. Please justify.

2. Line 47: Please provide also unadjusted results, in the form of Kaplan Meier/log rank testing, with rates overtime, i.e. at one year.

3. Line 49: it is unclear what this OR refers to re: intervention vs comparator. Is single antiplatelet better?

4. Line 91: I wonder if case-control studies can have power calculated, as you mention at this point. To the best of my knowledge, power is estimated only for RCTs.

5. Line 105: please add country.

6. Line 181 Sample size calculation: please refer to my previous comment.

7. Table 1: please include percentages for categorical data. Also provide exact p values, not just <0.05.

8. Line 309: please present results of univariate analysis,

9. Line 328: please check your statistics because I am not aware of a Chi2=0.

10. Line 336: the difference between the groups “70 (17.6%) in the single therapy group and 15 (12.6%)” could had been significant if your sample size was bigger, i.e. a type II error. Please comment.

Reviewer #2: Excellent article very well written. A retrospective observational study tracking >500 patients undergoing endovascular intervention.

Very tightly controlled design with robust analysis and good understanding of limitations of study design.

Couple of minor points - figures 2 and 3 have come across poorly. Figure 3 needs more description of the y axis - I assume this is cases?

Information from the STOPDAPT-2 study would be beneficial as this suggests that only 1 month of DAPT is required in DEB/DES in cardiac PCI. This may improve reader acceptance of this study.

Additionally, it is clear that in this study only approximately 1/3 of Drug eluting devices led to DAPT, which appears slightly unusual.

Was there any correlation between anatomical location and drug eluting status? This may be an interaction between covariates which may be of interest. I am sure this has been considered however.

6. PLOS authors have the option to publish the peer review history of their article (what does this mean?). If published, this will include your full peer review and any attached files.

Reviewer #1: Yes: Stavros Kakkos

Reviewer #2: No

---

## [Author Response · Author response to Decision Letter 0]

1 May 2020

* all page numbers refer to the tracked changes manuscript

Response: The formatting has been checked and updated. 

2. In ethics statement in the manuscript and in the online submission form, please provide additional information about the patient records used in your retrospective study. Specifically, please ensure that you have discussed whether all data were fully anonymized before you accessed them and/or whether the IRB or ethics committee waived the requirement for informed consent. If patients provided informed written consent to have data from their medical records used in research, please include this information.

Response: Thank you, this has been added. 

"This study was supported by the NIHR Biomedical Research Centre at

432 University Hospitals Bristol NHS Foundation Trust and the University of

433 Bristol. The views expressed in this publication are those of the authors and

434 not necessarily those of the NHS, the National Institute for Health Research or

435 the Department of Health and Social Care. GKA was supported by a Royal

436 College of Surgeons Research Fellowship. CPT was supported by Learning

437 and Research at North Bristol NHS trust.".

i) We note that you have provided funding information that is not currently declared in your Funding Statement. However, funding information should not appear in the Acknowledgments section or other areas of your manuscript. We will only publish funding information present in the Funding Statement section of the online submission form.

ii) Please remove any funding-related text from the manuscript and let us know how you would like to update your Funding Statement. Currently, your Funding Statement reads as follows:

"The author(s) received no specific funding for this work.".

Response: Thanks, the funding information in acknowledgments has now been moved to the correct place. 

Reviewer #1: General comments: The authors present a case controlled study on the possible association between DAPT and AFS. I have several comments outlined below.

1. I wonder what the reason was to study AFS as your primary endpoint because this would reduce the “power” of your study when looking at isolated limb outcomes. Please justify.

Response: There was some information on AFS choice in the primary endpoint section of the methods which has now been expanded. This now reads (line 172 page 9), “Primary endpoint: Amputation-free survival (composite of mortality plus major lower limb amputation defined as above ankle).2 We have performed patient focus group work as part of the run up for a randomised trial of single vs. dual antiplatelet therapy which confirmed this to be the most important patient centered outcome after treatment for chronic limb threatening ischaemia. It is also and is a widely accepted effectiveness outcome in chronic limb threatening ischaemia used in several major randomised trials.2”

2. Line 47: Please provide also unadjusted results, in the form of Kaplan Meier/log rank testing, with rates overtime, i.e. at one year.

Response: Thanks, this has been added.

3. Line 49: it is unclear what this OR refers to re: intervention vs comparator. Is single antiplatelet better?

Response: This has also been clarified by rewording the section to: “There was no difference in unadjusted amputation-free survival between patients with chronic limb threatening ischaemia taking single vs. dual antiplatelet therapy (69% vs. 74% respectively Log rank Chi2=0.1, p= .72). After adjusting for confounders, at 1 year there was also no significant difference in amputation-free survival between patients taking single vs. dual antiplatelet therapy [OR 0.8, 95% CI 0.5-1.2, p= .3].”

4. Line 91: I wonder if case-control studies can have power calculated, as you mention at this point. To the best of my knowledge, power is estimated only for RCTs.

Response: It is valid to do so, but is obviously nowhere near as powerful as for prospective studies. There are various arguments about different ways to do (for example here: Johnston KM, Lakzadeh P, Donato BMK, Szabo SM. Methods of sample size calculation in descriptive retrospective burden of illness studies. BMC Med Res Methodol. 2019;19(1):9. Published 2019 Jan 9. doi:10.1186/s12874-018-0657-9), it but we chose a ‘standard’ RCT type approach. This has been added to the discussion page 22 line 567, “Even though a sample size calculation was performed this is not as powerful for retrospective data as if the study was prospective. It is helpful to guide the number of patients included in the study but cannot be seen as definitive.”

5. Line 105: please add country.

Response: Thanks, this has been added. 

6. Line 181 Sample size calculation: please refer to my previous comment.

Response: Thanks, as above. 

7. Table 1: please include percentages for categorical data. Also provide exact p values, not just <0.05.

Response: This has been added. 

8. Line 309: please present results of univariate analysis,

Response: We didn’t run univariate analyses, just a multivariate model. We have now added the full model as a supplementary table (S1).

9. Line 328: please check your statistics because I am not aware of a Chi2=0.

Response: The Chi2 can equal zero when the expected values are equal to the observed values. 

10. Line 336: the difference between the groups “70 (17.6%) in the single therapy group and 15 (12.6%)” could had been significant if your sample size was bigger, i.e. a type II error. Please comment.

Response: Thanks, this should already have been commented on in the discussion. It has been added to the text page 22 line 567, “There is still the possibility of type 2 error in any of the comparison, especially as the groups become smaller.”

Reviewer #2: Excellent article very well written. A retrospective observational study tracking >500 patients undergoing endovascular intervention.

Very tightly controlled design with robust analysis and good understanding of limitations of study design.

Couple of minor points - figures 2 and 3 have come across poorly. Figure 3 needs more description of the y axis - I assume this is cases?

Response: Thanks, this was actually completely missing from 3. We’ve added an axis description to the title of each figure. 

Information from the STOPDAPT-2 study would be beneficial as this suggests that only 1 month of DAPT is required in DEB/DES in cardiac PCI. This may improve reader acceptance of this study.

Response: STOPDAPT-2 as been added to the discussion page 21 line 543. 

Additionally, it is clear that in this study only approximately 1/3 of Drug eluting devices led to DAPT, which appears slightly unusual.

Response: Yes, it wasn’t routinely used. This was also shown as common in our international survey of practice (reference 4)

Was there any correlation between anatomical location and drug eluting status? This may be an interaction between covariates which may be of interest. I am sure this has been considered however.

Response: This was not considered because it will exist by definition: DES are only routinely available for the SFA. It would also be off topic. 

Reviewer 3

An interesting and timely manuscript. I have a number of comments.

1. Any study looking at PAD patients is difficult and results tempered by the heterogeneity seen within the population – especially as the authors have included aorto-iliac / femoropopliteal and infrapopliteal endovascular revasculairsation in this dataset. Decision making around the role of antiplatelet therapy can be nuanced and often involves lesion type / endovascular result / run off / patient risk of bleeding – especially given a lack of antiplatelet therapy protocol within the department. To this end such data would be useful to allow further interpretation of the results. Specifically, eg TASC lesion criteria treated, run off – CFA / SFA / PFA disease of AI treatment and crural run off for FP treated lesions. Of more importance is risk of bleeding for an older and frail patient group as this may well have influenced decision making eg use of PPI / H2 antagonists at the time of revascularisation as a proxy for GI bleed risk or a HASBLEED 2 score. I am also surprised that allergy to aspirin / other antiplatelet therapies wasn’t an exclusion criterion. 

Response: Thanks, these are included in table 1 to the best extent available to us. We don’t have power to stratify results by TASC/anatomical location etc so doing so would be meaningless. Allergy would have been an exclusion by definition as it is a retrospective study and these patients were taking aspirin.

2. The authors admit that there is in built bias with regard to decision making around dual antiplatelet therapy – was there a difference between surgeons and radiologists. While managing the change in antiplatelet strategy is straightforward if the patient is an in patient what was the logistical arrangement for those patient attending as an out patient. Was the patient sent home with a prescription or was this left in the hands of the GP with a discharge letter – this is important as planned antiplatelet management does not equate to initiation of therapy / adherence. Adherence is a whole different ball game which this study would not be able to determine and has to be accepted as a weakness of the study. 

Response: This information has been added to the methods page 10 line 245, “The hospital prescription would last for 4-6 weeks and a summary would be given to the patient and GP for continuation.” Adherence/compliance was listed as a limitation in the discussion.

3. The trend of change in AP prescription does cloud the data and it may be that the use of DAP is better but this hasn’t been seen yet as the effect are seen out to 2 o r3 or 4 years. Could the authors comment on this.

Response: The latest patients were included in 2016 so that we had at least 2 years of follow up (median 29 months). If benefit hasn’t already been seen it would not be seen after this.

4. There are a number of assumptions around the sample size calculation which need to be mentioned in the discussion. 

Response: We used standard SSC methodology. It was based on other retrospective data, but often so is a SSC for a major RCT. If good data were available for a SSC the study would be pointless!

5. It is not well signposted in the manuscript that claudicants were excluded. Further is the data looking at per limb or per patient. Did the fact that a patient had two limbs treated alter the management of antiplatelet regime. Eg. If the ipisilateral limb was treated and patient put on DAP, and the the contra limb treated – was this patient then excluded as they were already on dual antiplatelet therapy? If the same patient was initially on a single AP agent and then after the second procedure put on DAP in which group were they classed? Would the data be cleaner if only those patients having one limb treated were included. 

Response: The title does state that we are interested in CLTI. It is also stated in each of the results subsection headings. 

 Contra limb; in the CLTI group there were all separate patients as well as limbs, i.e. there were no patients counted twice. To clarify this we have made several edits to the results. 

6. The authors need to define the comorbidities – ie. What is actually meant by ischaemic heart disease – by having detailed patients notes I am sure that they have strict definitions for inclusion.

Response: Thanks, we have added this information to the methods page 8 line 188.

7. Which variables were excluded with data missing of >15%. Which of the variable used had significant data missing eg>5%?

Response: This has been added to the results page 14 line 353, “Only two variables were excluded because of >15% missing data: smoking and the Wound Ischaemia and Foot infection score.” It was already listed as a limitation in the discussion.

---

## [Decision Letter · Decision Letter 1]

22 May 2020

Single versus dual antiplatelet therapy following peripheral arterial endovascular intervention for chronic limb threatening ischaemia: retrospective cohort study

PONE-D-20-04980R1

Dear Dr. Twine,

We are pleased to inform you that your manuscript has been judged scientifically suitable for publication and will be formally accepted for publication once it complies with all outstanding technical requirements.

With kind regards,

Athanasios Saratzis

Academic Editor

PLOS ONE

Additional Editor Comments (optional):

Reviewers' comments:

Reviewer's Responses to Questions

**Comments to the Author**

1. If the authors have adequately addressed your comments raised in a previous round of review and you feel that this manuscript is now acceptable for publication, you may indicate that here to bypass the “Comments to the Author” section, enter your conflict of interest statement in the “Confidential to Editor” section, and submit your "Accept" recommendation.

Reviewer #1: All comments have been addressed

Reviewer #2: All comments have been addressed

2. Is the manuscript technically sound, and do the data support the conclusions?

Reviewer #1: Yes

Reviewer #2: Yes

3. Has the statistical analysis been performed appropriately and rigorously? 

Reviewer #1: Yes

Reviewer #2: Yes

4. Have the authors made all data underlying the findings in their manuscript fully available?

Reviewer #1: Yes

Reviewer #2: Yes

5. Is the manuscript presented in an intelligible fashion and written in standard English?

Reviewer #1: Yes

Reviewer #2: Yes

6. Review Comments to the Author

Reviewer #1: None, all comments addressed

Reviewer #2: Thank you for the adjustments to this excellent paper.

This has made an improvement on an already sturdy paper.

7. PLOS authors have the option to publish the peer review history of their article (what does this mean?). If published, this will include your full peer review and any attached files.

Reviewer #1: Yes: Stavros K. Kakkos

Reviewer #2: No

---

## [Editor Report · Acceptance letter]

1 Jun 2020

PONE-D-20-04980R1 

Single versus dual antiplatelet therapy following peripheral arterial endovascular intervention for chronic limb threatening ischaemia: retrospective cohort study 

Dear Dr. Twine:

I am pleased to inform you that your manuscript has been deemed suitable for publication in PLOS ONE. Congratulations! Your manuscript is now with our production department. 

With kind regards,

on behalf of

Dr. Athanasios Saratzis 

Academic Editor

PLOS ONE